# Psychometric properties of the Portuguese version of the National Eye Institute Visual Function Questionnaire-25

**Ricardo Y. Abe**[1]*, **Felipe A. Medeiros**[2], **Milton Agrizzi Davi**[1], **Cecília Gonçalves**[1], **Matheus Bittencourt**[1], **Alicia Buffoni Roque**[1], **Júlia Boccato**[1], **Vital Paulino Costa**[1], **José Paulo Cabral Vasconcellos**[1]

**1** Department of Ophthalmology, University of Campinas, Campinas, Brazil, **2** Department of Ophthalmology, Duke University, Durham, North Carolina, United States of America

☉ These authors contributed equally to this work.
* ricardoabe85@yahoo.com.br

## Abstract

### Background

To investigate the psychometric properties of the Brazilian Portuguese version of the National Eye Institute Visual Function Questionnaire (NEI VFQ-25) questionnaire in a group of patients with different eye diseases.

### Methods

Cross-sectional study. All subjects completed the Portuguese version of the NEI VFQ-25 questionnaire. Another questionnaire containing a survey about clinical and demographics data was also applied. Rasch analysis was used to evaluate the psychometric properties of the NEI VFQ-25.

### Results

The study included 104 patients with cataract, 65 with glaucoma and 83 with age macular degeneration. Mean age was 70.7 ± 9.9 years, with 143 female (56.7%) and 109 male patients (43.2%). Mean visual acuity was 0.47 and 1.17 logMAR in the better and worse eye, respectively. According to Rasch analysis, seven items were found to misfit. Those items belonged to the following subscales: general health, social function, mental health, ocular pain and role limitations. The principal component analysis of the residuals showed that 55.5% of the variance was explained by the principal component. Eight items loaded positively onto the first contrast with a correlation higher than 0.4. These items belonged to the following subscales: near vision, distance vision, mental health and dependency. After excluding those items, we were able to isolate items from the NEI VFQ-25, related only to a visual functioning component. Finally, the principal component analysis from residuals of this revised version of the NEI VFQ-25 (items related to visual function) showed that the principal component explained 61.2% of the variance, showing no evidence of multidimensionality.

**Data Availability Statement:** We have no ethical or legal restrictions on sharing our data set and we have made it available within the Supporting Information files.

**Funding:** The authors received no specific funding for this work.

**Competing interests:** The authors have declared that no competing interests exist.

## Conclusions

The Portuguese version of the NEI VFQ-25 is not a unidimensional instrument. We were able to find items that belong to a different trait, possible related to a socio-emotional component. Thus, in order to obtain psychometrically valid constructs, both the visual functioning and socio-emotional components should be analyzed separately.

## Introduction

Quality of life (QoL) is a broad-ranging concept affected by an individual's physical health, psychological state, level of independence and social relationships.[1, 2] Within physical health, sense of sight is crucial to perform many routine daily activities. Therefore, changes in visual status can lead to functional impairment affecting directly QoL. Conventional clinical measures such as visual acuity, visual field assessment and fundus imaging may not fully capture the impact of disability related to eye diseases. Thus measurements of health-related QoL have been used to track outcomes for many eye diseases.[3, 4] Even though many so-called health-related QoL questionnaires can measure only a self-perceived health status, the importance of evaluating the outcomes of health care from the standpoint of the patient is now widely recognized.[5, 6]

Within patient-reported questionnaires, the National Eye Institute Visual Function Questionnaire (NEI VFQ-25) has been frequently used to assess QoL in ophthalmology research. [7–10] This questionnaire contains a set of 25 questions in 12 subscales designed to assess the dimensions of self-reported vision-target health status that are relevant for subjects with chronic eye diseases.[7] Items in the questionnaire require the subject to provide a response based on a Likert scale and methods for analyzing this type of data used in most previous studies attribute linear scores to each response and then sum up the scores for all different questions to obtain a single composite score. However, in order for a composite score to be meaningful is essential that all questions included in the scoring contribute to the measurement of a single underlying construct.[3, 11] For example, for the NEI VFQ-25 responses to be represented by a single score, its questions should all be measuring the same latent construct of visual functioning.

Rasch analysis is a method that can be used to investigate psychometric properties of questionnaires, such as dimensionality and reliability. Massof et al administered some items from the 52-NEI-VFQ to patients with low vision, applying Rasch model to estimate interval measurement scales from ordinal responses to items.[11] They found that Rasch analysis can offer an alternative to traditional scoring methods enabling one to estimate the latent variable of interest (visual function) and assess the performance of each item as a contributor to the final measurement. In a subsequent work, Marella and colleagues have suggested that the NEI VFQ-25 questionnaire does not seem to be unidimensional, and that the questionnaire items may actually be measuring two different underlying constructs, one related to visual functioning and another to socio-emotional status. This is important, as it would indicate that a single composite score is not appropriate to represent responses to this questionnaire. [12, 13] In addition to dimensionality, Rasch analysis can provide information about appropriateness of the response categories, measurement precision, and item fit to the construct.[14] Rasch analysis of the English version of the NEI VFQ-25 has also suggested that the subscales represented on the questionnaire would not be valid in their current format.[15]

As a widely used instrument to assess vision-related QoL, the NEI VFQ-25 has been translated into several different languages. When a questionnaire is translated into a new language, a linguistic validation is necessary but not sufficient unless the psychometric characteristics have been verified. Simao et al. introduced the Brazilian Portuguese version of the NEI VFQ-25 in 2008 and reported psychometrics properties comparable to the American original version.[16] However, the work of Simao et al reported only measures such as Cronbach's alpha and correlations among subscales. The reason Rasch analysis is well suited for demonstrating a translated version of an existing questionnaire has comparable items to the original is its sensitivity to differences in item difficulty.[17] Cronbach's alpha, for example, is insensitive to this, and can provide the same value for two questionnaires whose items differ in levels of difficulty. Thus, a proper validation of the Portuguese translation of the questionnaire would also benefit from a method such as Rasch analysis to better assess dimensionality and validity. To the best of our knowledge no study has yet applied the Rasch Analysis to the Portuguese version of the NEI VFQ-25 questionnaire. Thus, the purpose of current study is to investigate the psychometric properties of NEI VFQ-25 using Rasch analysis in a population of Brazilian patients with a variety of eye diseases.

## Materials and methods

This was a cross sectional study, evaluating patients with glaucoma, cataract and age-related macular degeneration (AMD) from the Hospital das Clínicas–University of Campinas–Brazil. This study was approved by the Ethics Committee of the University of Campinas and adhered to the tenets of the Declaration of Helsinki. Subjects underwent a comprehensive ophthalmic examination, including Snellen best corrected visual acuity, slit-lamp biomicroscopy, intraocular pressure (IOP) measurement using Goldmann applanation tonometry, gonioscopy and dilated fundoscopy examination using a 78-diopter lens. Subjects underwent standard automated perimetry, using the 24–2 Swedish Interactive Threshold Algorithm (SITA) Standard (Humphrey, Carl Zeiss Meditec, Inc., Dublin, CA, USA) and also retinal imaging with non-myd WX$^{3D}$ (Kowa, Japan).

Three different groups were investigated. Glaucoma patients were required to have repeatable (at least 2 consecutive) abnormal SAP results with corresponding glaucomatous optic nerve damage in at least one eye. An abnormal SAP result was defined as a pattern standard deviation with P < 0.05, and/or Glaucoma Hemifield Test results outside normal limits. Cataracts were classified according to the Lens Opacities Classification System III (LOCS III) based on findings from slit lamp examination.[18] For AMD, we applied the Clinical Age-Related Maculopathy Staging system, which divides patients into 5 mutually exclusive categories based on slit-lamp assessment of drusen, retinal pigment epithelial irregularities, geographic atrophy, retinal pigment epithelial detachment, and choroidal neovascularization.[19, 20]

### NEI VFQ-25 Questionnaire

Vision-related QoL was assessed by a Brazilian Portuguese version of the NEI VFQ-25 questionnaire.[7] This version was developed by Simao et al in 2008 and was initially tested for in a set of ophthalmic patients and healthy controls.[16] The NEI VFQ-25 consists of 25 questions measuring overall vision, difficulty with near-vision and distance activities, ocular pain, driving difficulties, limitations with peripheral vision and color vision, social functioning, role limitations, dependency and mental health symptoms. Rasch analysis was performed to obtain final estimates of "person measures," which can be used to express where each respondent falls on a linear scale representing the degree of impairment as measured by the NEI VFQ-25.[21, 22] Rasch analysis was performed using Andrich rating-scale models to obtain the estimates of

the required ability of each item, perceived ability of each subject, and the thresholds for each response category.[14] The unit of those estimated measures is called a logit (log-odds unit), which is calculated as the log-odds ratio of the probability that a participant will select a particular rating category in an item over 1 minus the same probability. The logit values place patients according to their abilities and items according to their difficulties on the same linear interval scale. [23]

Person and item measures were examined for fit to the Rasch model using infit and outfit item fit statistics.[8] To test the hypothesis that the NEI VFQ- 25 measures a single underlying construct, we initially evaluated the fit statistics, which were recorded as mean square standardized residuals (MNSQ); The fit of the Rasch model was evaluated with the infit and outfit statistics. Values between 0.7 and 1.3 are considered acceptable for MNSQ values of infit and outfit.[24] After checking fit statistics, we conducted a principal components analysis of the residuals (difference between the observed and expected responses).[25] Data are considered unidimensional if most of the variance is explained by the principal component and there is no significant explanation of the residual variance by the contrasts to the principal component. In general, to be considered unidimensional, the variance of the principal component should be >60%.[8, 25] Furthermore, the unexplained variance by the contrasts should be <2 Eingenvalue units.[25]

We also evaluated differential item functioning, which assesses whether the items have different meanings for different groups in the sample. The raw differences in item calibration between groups were examined to identify differential item functioning. The differential item functioning was considered absent if it was less than 0.50 logits, minimal but probably inconsequential if it ranged between 0.50 and 1.0 logits, and notable if it was >1.0 logit.[12, 26]

The person separation index is the ratio of the variance in the person measures for the sample to the average error in estimating these measures. It is a measure of how broadly the persons could be distinguished into statistically distinct levels. The person separation reliability coefficient describes the reliability of the scale to discriminate between the persons of different abilities.[12] A person separation index of $\geq 2.0$ or a reliability value of $\geq 0.8$ represents the minimum acceptable level of separation.[12, 24]

## Demographic, clinical and socio-economic variables

Socio-economic questionnaires were also administered along with the NEI VFQ-25 to all patients. These questionnaires contained a survey about demographics, history of ocular and medical conditions, marital status, degree of education and income. For comorbidities, we investigated the presence or history of the following conditions: diabetes mellitus, arthritis, high blood pressure, heart disease, depression, asthma, and cancers. A simple summation score was used to create a comorbidity index.[27] As these variables could potentially affect patient perceptions about vision-related QOL, they were included as covariates factors to investigate their association with the final Rasch-calibrated NEI VFQ-25 scores. These variables were categorized for inclusion in the univariate and multivariate models as race (African American [yes/no]), employment (yes/no), marital status (married [yes/no]), degree of education (at least high school degree [yes/no]) and income (less than $25,000/year [yes/no]). Visual acuity was measured using an Early Treatment Diabetic Retinopathy (ETDRS) chart and logMAR measurements were used in the analyses evaluating better and worse eye.[28] For patients with visual acuity measures of "counting fingers" (CF), "hand motion", "light perception" and "no light perception" (NLP), we converted into quantitative measurements such as logMAR, as suggested by Schulze-Bonsel. [28]

## Statistical analysis

Descriptive statistics included mean and standard deviation for normally distributed variables. We investigated the relationship between final Rasch-calibrated NEI VFQ-25 scores with socio-economic and clinical variables (gender, race, education, income, marital status, visual acuity in logMAR, presence of low vision and mean deviation from standard automated perimetry) using a linear regression model. Variables with P value < 0.2 were included in the final multivariable linear regression model. Statistical analyses were performed using Winsteps 3.81.0 (Chicago, IL) and STATA v. 13 (StataCorp, College Station, TX). The alpha level (type I error) was set at 0.05.

## Results

The study included 104 patients with cataract, 65 with glaucoma and 83 with AMD. Table 1 presents demographic variables of the studied population. Mean age was 70.7 ± 9.9 years, with

**Table 1. Demographic characteristics of all patients included in the study.**

| Parameters | Total subjects (n = 252) |
|---|---|
| **Age (years)** | |
| Mean ± SD | 70.7± 9.9 |
| Range | 30 to 103 |
| **Gender, n (%)** | |
| Male | 109 (43.2%) |
| Female | 143 (56.7%) |
| **Race, n (%)** | |
| Caucasian | 204 (81.2%) |
| African-American | 44 (17.5%) |
| **Job status (%)** | |
| Employed | 44 (18.1%) |
| Unnenployed | 19 (7.8%) |
| Retired | 180 (74.0%) |
| **Marital status (%)** | |
| Married | 119 (69.5%) |
| Single | 25 (14.6%) |
| Widowed | 14 (8.1%) |
| Divorced | 13 (7.6%) |
| **Education (%)** | |
| Illiterate | 2 (0.8%) |
| Elementary school | 137 (56.6%) |
| High school degree | 58 (23.9%) |
| College degree | 8 (3.2%) |
| **Income per month (%)** | |
| Lower than US$414.00 | 81 (48.2%) |
| Between US$414.00 and US$2,073.00 | 68 (40.4%) |
| Between US$2,073.00 and US$4,147.00 | 17 (10.1%) |
| Higher than US$4,147.00 | 2 (1.1%) |
| **Comorbidity Index (%)** | |
| Zero | 71 (28.9%) |
| One | 109 (44.4%) |
| Two | 58 (23.6%) |
| Three | 6 (2.4%) |

143 female (56.7%) and 109 male patients (43.2%). Most of them were retired (74%). Table 2 describes the clinical variables of the patients. Mean visual acuity in the better eye was 0.47 log-MAR and 1.17 logMAR in worse eye. There were 62 patients (24.6%) with low vision (counting fingers, hand motion, light perception or loss of light perception in one or both eyes).

## Rasch analysis

Results of Rasch analysis are shown in Table 3. Five items (Q4, Q11, Q17, Q19 and Q25) were found to misfit (from subscales: social function, mental health, ocular pain and role limitations) with infit and/or outfit mean scores >1.3 and <0.7 (S1 Fig) Principal components analyses of the residuals from Rasch analysis can also be used to check the assumption of unidimensionality.[29] In order to determine whether the assumption of unidimensionality is valid, the variance explained by the Rasch factor (the underlying construct) should be 4 times greater than that of the first principal component in the residuals and the variance explained by the Rasch factor should be greater than 60%.[30]

For the current work, the principal component analysis of the residuals showed that the variance explained by the principal component was comparable for empirical calculation (55.5%) and by the model (56.1%) (S2 Fig). This suggests that the questionnaire was not unidimensional. Moreover, the unexplained variance explained by the first contrast was 2.81 eigenvalue units and the second contrast was 2.04 eigenvalue units with no further contrasts exceeding 2.0 eigenvalue units. These findings suggested the presence of a second dimension in the scale. Eight items loaded (correlation>0.4) positively onto the first contrast and belonged to: near vision (Q5), distance vision (Q8 and Q14), mental health (Q21and Q22) and dependency (Q20, Q23 and Q24) (S3 Fig). This suggests that these eight items cannot be grouped with other items in the scale to measure a single latent trait (visual functioning). These items are probably related to a secondary component (probably, a social-emotional component). Of note, in the current sample, 187 patients (74.4%) answered that they were not currently driving (Q15). Within this group, 158 patients (84.5%) reported that they never had driven (Q15a). Therefore questions related to driving were not assessed in the Rasch Analysis due to missing data.

Differential item functioning was tested for some of the variables from Table 1 and Table 2, such as: age, gender, race, job status, marital status, education, level of income, low vision and type of eye disease (cataract, glaucoma and AMD). There was no differential item functioning for any of the variables mentioned. These results suggest that items could be interpreted similarly across subgroups of the sample.

**Table 2. Clinical characteristics of all patients included in the study.**

| Parameters | Total subjects (n = 252) |
|---|---|
| LogMar Visual acuity (better eye) | |
| Mean ± SD | 0.47 ± 0.39 |
| LogMar Visual acuity (worse eye) | |
| Mean ± SD | 1.17 ± 0.74 |
| SAP MD from glaucoma (better eye) (dB) | |
| Mean ± SD | -4.26 ± 3.85 |
| SAP MD OS from glaucoma (worse eye) (dB) | |
| Mean ± SD | -10.77 ± 9.38 |
| Low vision, n (%) | 62 (24.6%) |

SD: standard deviation; SAP: standard automated perimetry; MD: mean deviation; OD: right eye; OS: left eye.

**Table 3. Fit Statistics using Rasch Analysis with respective Items and Subscales from National Eye Institute Visual Function Questionnaire (NEI VFQ-25).**

| Questions | Items | Subscales | Measure | Infit MNSQ | Outfit MNSQ |
|---|---|---|---|---|---|
| Q1 | General health | General health | 0.03 | 0.98 | 1.28 |
| Q2 | General vision | General vision | 0.02 | 0.62 | 0.77 |
| Q3 | Worry about eyesight | Mental health | 0.05 | 1.20 | 1.24 |
| Q4 | Pain around eyes | Ocular pain | -0.02 | 1.58 | 2.03 |
| Q5 | Reading normal newsprint | Near vision | 0.03 | 0.93 | 0.93 |
| Q6 | Seeing well up close | Near vision | 0.01 | 0.88 | 0.83 |
| Q7 | Finding objects on crowded shelf | Near vision | -0.01 | 0.79 | 0.73 |
| Q8 | Street signs | Distance vision | 0.01 | 0.91 | 0.92 |
| Q9 | Going downstairs at night | Distance vision | 0.00 | 0.79 | 0.75 |
| Q10 | Seeing objects off to side | Peripheral vision | -0.01 | 0.72 | 0.71 |
| Q11 | Seeing how people react | Social function | -0.04 | 0.90 | 0.61 |
| Q12 | Matching clothes | Color vision | -0.04 | 0.85 | 0.84 |
| Q13 | Visiting others | Social function | -0.02 | 1.07 | 0.87 |
| Q14 | Going out to movies/plays | Distance vision | 0.00 | 1.19 | 1.24 |
| Q17 | Accomplish less | Role limitations | 0.01 | 1.40 | 1.41 |
| Q18 | Limited endurance | Role limitations | 0.01 | 0.88 | 0.83 |
| Q19 | Amount of time in pain | Ocular pain | 0.00 | 1.65 | 1.75 |
| Q20 | Stay home most of the time | Dependency | -0.01 | 0.89 | 0.83 |
| Q21 | Frustrated | Mental health | 0.01 | 1.06 | 1.02 |
| Q22 | No control | Mental health | 0.00 | 1.01 | 0.92 |
| Q23 | Rely too much on others' words | Dependency | -0.02 | 1.07 | 0.87 |
| Q24 | Need much help from others | Dependency | -0.01 | 0.96 | 0.87 |
| Q25 | Embarrassment | Mental health | -0.02 | 1.30 | 0.99 |

MNSQ: mean square

After excluding items that were considered misfitted (Q4, Q11, Q17, Q19 and Q25) and also those items with high loadings on the principal component analysis of the residuals, such as: near vision (Q5), distance vision (Q8 and Q14), mental health (Q21and Q22) and dependency (Q20, Q23 and Q24), we were able to isolate items from the NEI VFQ-25, related only to the first component (probably, a visual function component). After running new analysis with items related visual function, we found that items Q1 and Q3 had values for infit and outfit >1.3 (S4 Fig). After excluding those items, we performed new Rasch Analysis and according to Table 4, no items were misfitted. We also performed a principal component analysis of the residuals of the revised version of the NEI VFQ-25 (items related to visual function and socioemotional component). The final variance of the principal component was 61.2% and the unexplained variance by the contrasts is 1.49 eingenvalue units, showing no evidence of multidimensionality (Table 5). The mean (± SD) of the person measures was 0.05 ± 0.02 logits. In Fig 1, we showed the Wright item-person maps of the revised version of the NEI VFQ-25 (only items related to visual function). The separation index for person measures was 2.17, with reliability of 0.83. We also reported the psychometric properties of the socioemotional component of the NEI VFQ-25 (Table 5).

We also investigated the association between demographic and clinical variables with the final scores of the revised version of the NEI VFQ-25. Within the clinical and demographic variables, there was a statistical relationship with the Rasch-calibrated scores in NEI VFQ-25 for the following variables in univariable models: visual acuity in the better eye (P<0.001), visual acuity in the worse eye (P<0.001), patients with low vision (P<0.001), gender

**Table 4. Fit Statistics from Rasch Analysis with respective Items and Subscales using the only item related to visual function of the National Eye Institute Visual Function Questionnaire (NEI VFQ-25).**

| Questions | Items | Subscales | Measure | Infit MNSQ | Outfit MNSQ |
|---|---|---|---|---|---|
| Q2 | General vision | General vision | 0.04 | 1.01 | 1.25 |
| Q6 | Seeing well up close | Near vision | 0.03 | 1.03 | 0.96 |
| Q7 | Finding objects on crowded shelf | Near vision | -0.01 | 0.92 | 0.80 |
| Q9 | Going downstairs at night | Distance vision | 0.00 | 0.98 | 0.88 |
| Q10 | Seeing objects off to side | Peripheral vision | -0.01 | 0.77 | 0.68 |
| Q12 | Matching clothes | Color vision | -0.04 | 0.97 | 0.92 |
| Q13 | Visiting others | Social function | -0.03 | 1.29 | 0.93 |
| Q18 | Limited endurance | Role Limitations | 0.02 | 1.28 | 1.15 |

MNSQ: mean square

(P<0.001), marital status (P = 0.001), employment status (P = 0.019), education level (P<0.001) and comorbidity index (P = 0.003). In a multivariable analysis, only 2 variables remained statistically significant: visual acuity in the better eye (P<0.001) and education level (P = 0.002) (S1 Table).

## Discussion

In the current study, we investigated the psychometric properties of the Brazilian Portuguese version of the NEI VFQ-25 using Rasch analysis. Our results showed that the NEI VFQ-25 does not seem to be an unidimensional instrument; that is, it does not measure a single latent construct (quality of life related to visual function).[11] Although most items on the NEI VFQ-25 tap the construct of visual functioning, our results indicated that other items belonged to a different construct, namely socio-emotional component, corroborating findings from previous studies.[12, 13]

Unidimensionality of an instrument can be assessed by examining the fit statistics and principal component analysis of the residuals. Ideally items should have MNSQ values between 0.7 and 1.3. Items with MNSQ lower than 0.7 suggest a high level of predictability in the responses, indicating redundancy, [24] whereas values higher than 1.3 show an unacceptable level of noise in the responses. In total, seven items (Q1, Q3, Q4, Q11, Q17, Q19 and Q25) were found to misfit. Those items belonged to the subscales of general health, social function, mental health, ocular pain and role limitations

**Table 5. Rasch Analysis Fit Statistics of the Visual Function and Socioemotional Components from the National Eye Institute Visual Function Questionnaire (NEI VFQ 25).**

| Components | Visual Function | Socioemotional |
|---|---|---|
| Items in Scale (n) | 8 | 8 |
| Misfitting Items (n) | None | None |
| Person Separation Index | 2.17 | 1.89 |
| Person Separation Reliability (logits) | 0.83 | 0.78 |
| Mean Person Measure (logits) | 0.05 | 0.02 |
| Final variance of Principal Component (%) | 61.20 | 60.8 |

```
MEASURE      PERSON - MAP - ITEM
                    <more>|<rare>
       1                  +
                          |
                          |
                          |
                          |
                          |
                          |
                          |
                          |
                          |
                          |
                          |
                          |
                          |
               .##  T|
               .##   |
         .##########  S|
          .#########   |T
         .##########  M|S  Q18      Q2       Q6
       0      ########  S+M  Q10      Q7       Q9
                .####   |S  Q12      Q13
                  .#  T|T
                   .   |
                   .   |
                       |
                       |
                       |
                       |
                       |
                       |
                       |
                       |
                       |
                       |
      -1          .   +
                <less>|<freq>
     EACH "#" IS 5: EACH "." IS 1 TO 4
```

**Fig 1. Wright item-person maps related to visual function of the revised version of the National Eye Institute Visual Function Questionnaire (NEI VFQ-25).** The left-hand column locates the person ability measures along the variable. The right-hand column locates the item difficulty measures along the variable.

In addition to that, we also need to exam the principal component analysis of the residuals as a second test for unidimensionality. A high level of variance accounted for by the principal component leads to a low likelihood of additional components; a variance of 60% or greater is considered good. In the current study, the principal component analysis of the residuals showed that the variance explained by the principal component was 55.5%. Moreover, the unexplained variance explained by the first contrast was 2.81 eigenvalue units. The first contrast in the residuals reports whether there are patterns within variance that are unexplained by the principal component, which suggests a second construct is being measured.[13] According to previous studies, the current study applied the criterion that the contrast should have an eigenvalue higher than 2.0 to be considered evidence of a second construct because this would be greater than the magnitude seen with random data. Thus, our analysis showing the first contrast with a 2.81 eigenvalue units, suggests that the Brazilian Portuguese version of the NEI VFQ-25 was not unidimensional.

The loading of items onto the contrasts allows identification of which items tap different constructs. In our analysis, eight items loaded positively onto the first contrast with a correlation higher than 0.4. These items belonged to the following subscales: near vision (Q5), distance vision (Q8 and Q14), mental health (Q21and Q22) and dependency (Q20, Q23 and Q24),. This suggests that these eight items cannot be grouped with other items in the scale to measure a single latent construct, such as QoL related to visual function.

We were able to isolate items from the NEI VFQ-25, related only to a visual function component, after excluding items that were considered misfitted and also those items with high loadings on the principal component analysis of the residuals. When we re-examined the fit statistics of this revised version of the NEI VFQ-25, two items still presented infit and outfit values higher than 1.3 (Q1 and Q3). After excluding those items, new analysis was performed and no items were misfitted (Table 4). Moreover, the final variance of the principal component was 61.2% and the unexplained variance by the first contrast was 1.49 eingenvalue units (Table 5). These results suggest that this revised version of the NEI VFQ-25 showed no evidence of multidimensionality.

Simao et al used a "Factor analysis" and concluded that almost all subscales of NEI VFQ-25 belong to the same underlying dimension. However, careful analysis of their data suggests some evidence of multidimensionality.[16] For example, they showed that most of the subscales from the Portuguese version of the NEI VFQ-25 were influenced by central vision correlated with the first factor, while the "General vision", "Ocular pain" and "Peripheral vision" subscales were included in a second factor.[16] We were able to find a second construct more related to a socio-emotional component formed by subscales such as: "General health", "mental health", "role limitations" and "dependency", in contrast to central and peripheral vision constructs as highlighted in the previous study. This difference might be due to application of different types of analysis (Rasch as opposed to Factor Analysis). When evaluating an instrument with the Rasch model, more fundamental evidence may be provided to justify the use of scale scores on an interval level. Distances on the scales developed by the Factor Analysis approach are interpreted as equal over the full range of the scale.[31] The scale is treated as an interval scale based on ordinal level item scoring. In fact, Pesudovs et al investigated the psychometric properties of the NEI VFQ-25 with Rasch analysis in a group of patients with cataract and found that several subscales were not psychometrically sound. They concluded that the NEI VFQ-25 as an overall measure was flawed by multidimensionality.[13]

Marella et al performed a similar investigation with a group of low vision patients and found that the NEI VFQ-25 is a better performing instrument when divided into two different scales, corroborating the findings of our study. [12] In their study, 3 items were misfit (general health, pain around eyes, driving in difficult conditions) with infit mean scores higher than 1.3. The principal component analysis of the item residuals revealed that eight items loaded (correlation higher than 0.4) positively onto the first contrast and belonged to dependency (three items), mental health (three items), and role limitations (two items) subscales.

Another important characteristic of a good instrument is that items function similarly for persons at the same level of ability. Differential item functioning was tested for the following variables: age, sex, race, job status, marital status, education, level of income, low vision and type of eye disease (cataract, glaucoma and AMD). Differential item functioning occurs when subgroups of people with comparable levels of ability respond differently to an item, which implies a response to some characteristic other than item difficulty. We were not able to find evidence of differential item functioning for any of the variables mentioned. Thus, our results suggest that items from the Portuguese version of the NEI VFQ-25 could be interpreted similarly across subgroups of the sample, including different eye diseases, such as cataract, glaucoma and AMD.

We found that worse visual acuity and patients with lower education level had lower Rasch-calibrated NEI VFQ-25 scores. Even though patients with AMD had lower Rasch-calibrated scores of NEI VFQ-25 compared to cataract and glaucoma patients, when adjusting for visual acuity, the correlation with different types of eye disease in the multivariable analysis was not statistically significant, implying that visual acuity may be a better predictor for vision related QoL in comparison to the underlying cause of the vision loss. In fact, associations between worse visual acuity and QoL have already been demonstrated.[7, 32] Moreover, previous work have suggested that poor educated patients might have higher levels of emotional distress (including depression, anxiety, and anger) and physical distress (including aches and pains and malaise), which could influence the responses of the QoL questionnaire.[33]

The current study has limitations. Even though Rasch analysis is becoming the gold standard for scoring patient-reported outcome measures in ophthalmology, a multilevel model that allows simultaneous analysis of different dimensions in a multidimensional instrument could also be used.[8, 10, 34] Our sample consisted of patients with cataract, glaucoma and AMD. Thus, future studies should investigate psychometric validity of the Rasch calibrated version of the NEI VFQ-25 in a sample with more varied range of eye diseases.

## Conclusion

Our findings indicate that the Brazilian Portuguese version of the NEI VFQ-25 is not psychometrically optimal for assessing QoL related only to visual function. Rather, we found a second trait, described as a socioemotional component from results of the Rasch analysis. Thus, in order to obtain psychometrically valid constructs, both components with their respective subscales and items (visual functioning and socioemotional) should be analyzed separately. Future studies in Brazil including patients with different eye diseases are needed to substantiate our findings and evaluate the sensitivity of this calibrated version of the NEI VFQ-25.

## Supporting information

**S1 Fig. Item fit statistics for the National Eye Institute Visual Function Questionnaire-25.** (PNG)

**S2 Fig. Principal component analysis of the residuals.**
(PNG)

**S3 Fig. Corresponding loadings for the first contrast.**
(PNG)

**S4 Fig. Item fit statistics for revised version of the National Eye Institute Visual Function Questionnaire-25.**
(PNG)

**S1 Table. Univariate and Multivariate regression analysis between demographic and clinical variables and final scores of the revised version of the National Eye Institute Visual Function Questionnaire-25.**
(XLSX)

## Author Contributions

**Conceptualization:** Ricardo Y. Abe, Vital Paulino Costa, José Paulo Cabral Vasconcellos.

**Data curation:** Ricardo Y. Abe, Felipe A. Medeiros, Milton Agrizzi Davi, Cecília Gonçalves, Matheus Bittencourt, Alicia Buffoni Roque, Júlia Boccato, José Paulo Cabral Vasconcellos.

**Formal analysis:** Ricardo Y. Abe, Felipe A. Medeiros, Cecília Gonçalves.

**Investigation:** Ricardo Y. Abe, Felipe A. Medeiros, Milton Agrizzi Davi, Matheus Bittencourt, Alicia Buffoni Roque, Júlia Boccato.

**Methodology:** Ricardo Y. Abe, Cecília Gonçalves.

**Project administration:** Ricardo Y. Abe, Milton Agrizzi Davi, Cecília Gonçalves.

**Software:** Ricardo Y. Abe, Felipe A. Medeiros.

**Supervision:** Felipe A. Medeiros.

**Validation:** Ricardo Y. Abe, Felipe A. Medeiros.

**Visualization:** Ricardo Y. Abe, Vital Paulino Costa.

**Writing – original draft:** Ricardo Y. Abe, Vital Paulino Costa, José Paulo Cabral Vasconcellos.

**Writing – review & editing:** Ricardo Y. Abe, Felipe A. Medeiros, Vital Paulino Costa, José Paulo Cabral Vasconcellos.

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
