## [Decision Letter · Decision Letter 0]

12 Sep 2019

PONE-D-19-19847

Psychometric Properties of the Portuguese Version of the National Eye Institute Visual Function Questionnaire-25

PLOS ONE

Dear Dr. Abe,

Thank you for submitting your manuscript to PLOS ONE. After careful consideration, we feel that it has merit but does not fully meet PLOS ONE’s publication criteria as it currently stands. Therefore, we invite you to submit a revised version of the manuscript that fully addresses all the points raised during the review process in a concise manner.

We would appreciate receiving your revised manuscript by Oct 27 2019 11:59PM. To enhance the reproducibility of your results, we recommend that if applicable you deposit your laboratory protocols in protocols.io, where a protocol can be assigned its own identifier (DOI) such that it can be cited independently in the future. For instructions see: http://journals.plos.org/plosone/s/submission-guidelines#loc-laboratory-protocols

We look forward to receiving your revised manuscript.

Kind regards,

Stefan Hoefer

Academic Editor

PLOS ONE

Journal Requirements:

2. Thank you for stating the following financial disclosure: No

Please provide an amended Funding Statement that declares *all* the funding or sources of support received during this specific study (whether external or internal to your organization) as detailed online in our guide for authors at http://journals.plos.org/plosone/s/submit-now.  Please state what role the funders took in the study.  If any authors received a salary from any of your funders, please state which authors and which funder. If the funders had no role, please state: "The funders had no role in study design, data collection and analysis, decision to publish, or preparation of the manuscript."

4. We noticed you have some minor occurrence of overlapping text in the Discussion with the following previous publication(s), which needs to be addressed:

Prieto, Luis, Jordi Alonso, and Rosa Lamarca. "Classical test theory versus Rasch analysis for quality of life questionnaire reduction." Health and Quality of Life Outcomes 1.1 (2003): 27.

Pesudovs, Konrad, et al. "Remediating serious flaws in the National eye Institute visual function questionnaire." Journal of Cataract & Refractive Surgery 36.5 (2010): 718-732.

The text that needs to be addressed occurs in the third paragraph of the Discussion and begins, "A high level of variance accounted for..." and in the two sentences beginning, "When evaluating an instrument with the Rasch model..."

In your revision ensure you cite all your sources (including your own works), and quote or rephrase any duplicated text outside the methods section.

Reviewers' comments:

Reviewer's Responses to Questions

**Comments to the Author**

1. Is the manuscript technically sound, and do the data support the conclusions?

Reviewer #1: Yes

Reviewer #2: Yes

Reviewer #3: Yes

Reviewer #4: Yes

2. Has the statistical analysis been performed appropriately and rigorously? 

Reviewer #1: Yes

Reviewer #2: Yes

Reviewer #3: Yes

Reviewer #4: Yes

3. Have the authors made all data underlying the findings in their manuscript fully available?

Reviewer #1: Yes

Reviewer #2: Yes

Reviewer #3: Yes

Reviewer #4: Yes

4. Is the manuscript presented in an intelligible fashion and written in standard English?

Reviewer #1: Yes

Reviewer #2: No

Reviewer #3: Yes

Reviewer #4: Yes

5. Review Comments to the Author

Reviewer #1: The authors investigate the psychometric properties of the Brazilian Portuguese version of the National Eye Institute Visual Function Questionnaire (NEI VFQ-25) in a group of patients with different eye diseases. This is an important topic, as no study has yet applied Rasch Analysis to the Brazilian Portuguese version of the NEI VFQ-25. The study is well designed, executed and reported.

However, I have a few comments.

1. Material and Methods: Adding some subheading could improve flow and readability for the reader.

2. Not all readers are familiar with the NEI VFQ-25. Could you please provide more information about the NEI VFQ-25 (how initially developed, how many response options, which scaling [Likert scaling?], subscales)?

3. Targeting: Did you inspect the person-item map and calculated the difference between item and person means before revising the questionnaire? Knowing if the difficulty of the items adequately targets the ability of the sample provides valuable information.

4. Figure 2: The mean person achievement measure is much lower than the mean item difficulty, suggesting a lack of items at the low-ability end. Could you please explain and describe that? This finding is completely counter-intuitive as most questionnaires struggle with items that are too easy, in particular in questionnaires trying to capture VRQOL.

5. Table 2: Does the SAP MD values only refer to glaucoma patients or to the overall sample?

6. Could you pleaes add more information on visual ability of the participants? Please report n/% of the sample in the categories none/mild/moderate/severe visual impairment.

7. Line 252: The authors say, that four items were found to misfit with Infit and/or Outfit MNSQ values <0.7 or >1.3. In Table 3 Infit and Outfit MNSQ values for Q24 are within these values, so why was this item found to misfit? Same for Q25.

8. Line 226: The authors say, that they investigated the relationship between final Rasch-calibrated NEI VFQ-25 scores with socioeconomic variables using a linear regression model. Could you please provide the results in a table in the supplement to see raw and adjusted effects?

9. Most of the readers might not be familiar with Rasch analysis, so please explain in more detail the factor analysis/factor loading you used to assess unidimensionality of the scale. Otherwise it might not be easy to follow why items Q1, Q3, Q21, Q22, Q17, A18, Q20 and Q 23 were excluded?

10. Please cite additional literature which also reports the NEI-VFQ to be multi-dimensional.

11. Line 39: Typing error/ double word: “National Eye Institute Visual Questionnaire (NEI VFQ-25) questionnaire”

Reviewer #2: The purpose of this manuscript is to investigate the psychometric properties of the Brazilian Portuguese version of the National Eye Institute Visual Function Questionnaire (NEI VFQ-25) questionnaire in a group of patients with different eye diseases by Rasch analysis. However, The authers found that Portuguese version of the NEI VFQ-25 is not a unidimensional instrument in measuring psychometric properties. Although they suggest that analyzing both visual function and socio-emotional components separately may be a valid method, they did not test the validity of this method. In addition, some mistakes in grammar was found in the manuscript.

Reviewer #3: In this paper, Abe et al. investigated the psychometric properties of the Brazilian Portuguese version of the National Eye Institute Visual Function Questionnaire-25. They found the Portuguese version of NEI VFQ-25 is not a unidimensional instrument, and suggested the visual functioning and socio-emotional components should be analyzed separately. In general, the paper is well written and in a good quality. The study is well designed, the data are convincing, the analysis is cogent.

Specific comments follow:

1. In the methods part, the author may want to elaborate how the NEI VFQ-25 questionnaires were conducted. Were they self-administered or interviewer-administered?

2. In the methods part (line 206), “Socio-economic questionnaires were also administered along with the NEI VFQ-25 to all patients.” A sample of this questionnaire should be attached to this manuscript, preferably as supplementary materials.

3. In the results part (line 252), “Four items (Q4, Q19, Q24 and Q25) were found to misfit (from subscales: general health, mental health, ocular pain and role limitations) with infit and/or outfit mean scores >1.3.”. However, from Table 3, the infit MNSQ and outfit MNSQ of Q24 are 0.96 and 0.87 respectively, and are both less than 1.3.

4. As authors wrote in the intro (line 110-115), “Marella and colleagues have suggested that the NEI VFQ-25 questionnaire does not seem to be unidimensional, and that the questionnaire items may actually be measuring two different underlying constructs, one related to visual functioning and another to socio-emotional status.” It would be necessary for the authors to include the comparison between the findings from the present study with Marella et al.’s result.

5. It is surprising to see that the items authors found misfitting in this study (namely Q4, Q19, Q24 and Q25) were very much different from the misfitting questions Pesudovs et al. found in their study (PMID: 20457362). Authors need to provide explanations for this difference.

Reviewer #4: This is an interesting finding. The authors investigated the psychometric properties of NEI VFQ-25 using Rasch analysis in a population of Brazilian patients with a variety of eye diseases. They found that the Brazilian Portuguese version of the NEI VFQ-25 is not psychometrically optimal for assessing QoL related only to visual function. Moreover, they also observed a second trait, described as a socioemotional component from results of the Rasch analysis. Overall, the authors have described the experimental design and knowledge gaps to be generally filled with this study. The results have been generally presented in a reasonable and expected approach with solid statistical analysis.

6. PLOS authors have the option to publish the peer review history of their article (what does this mean?). If published, this will include your full peer review and any attached files.

Reviewer #1: No

Reviewer #2: Yes: Hongbing Zhang

Reviewer #3: No

Reviewer #4: No

---

## [Author Response · Author response to Decision Letter 0]

29 Sep 2019

We have addressed all comments and suggestions made by reviewers. Details are available in the point-by-point letter response attached.

---

## [Decision Letter · Decision Letter 1]

1 Nov 2019

PONE-D-19-19847R1

Psychometric Properties of the Portuguese Version of the National Eye Institute Visual Function Questionnaire-25

PLOS ONE

Dear Dr. Abe,

Thank you for submitting your manuscript to PLOS ONE. After careful consideration, we feel that it has merit but does not fully meet PLOS ONE’s publication criteria as it currently stands. Therefore, we invite you to submit a revised version of the manuscript that addresses the points raised during the review process.

We would appreciate receiving your revised manuscript by Dec 16 2019 11:59PM. To enhance the reproducibility of your results, we recommend that if applicable you deposit your laboratory protocols in protocols.io, where a protocol can be assigned its own identifier (DOI) such that it can be cited independently in the future. For instructions see: http://journals.plos.org/plosone/s/submission-guidelines#loc-laboratory-protocols

We look forward to receiving your revised manuscript.

Kind regards,

Stefan Hoefer

Academic Editor

PLOS ONE

Additional Editor Comments (if provided):

I have read the revised version of the manuscript myself and believe you have addressed most of the comments of the previews review well. New comments regarding the revised version have been raised. Please try to respond by either commenting or correcting in the manuscript the issues discussed by the reviewer.

Reviewers' comments:

Reviewer's Responses to Questions

**Comments to the Author**

1. If the authors have adequately addressed your comments raised in a previous round of review and you feel that this manuscript is now acceptable for publication, you may indicate that here to bypass the “Comments to the Author” section, enter your conflict of interest statement in the “Confidential to Editor” section, and submit your "Accept" recommendation.

Reviewer #2: (No Response)

Reviewer #3: All comments have been addressed

Reviewer #4: All comments have been addressed

Reviewer #5: All comments have been addressed

2. Is the manuscript technically sound, and do the data support the conclusions?

Reviewer #2: Partly

Reviewer #3: Yes

Reviewer #4: Yes

Reviewer #5: Yes

3. Has the statistical analysis been performed appropriately and rigorously? 

Reviewer #2: No

Reviewer #3: Yes

Reviewer #4: Yes

Reviewer #5: Yes

4. Have the authors made all data underlying the findings in their manuscript fully available?

Reviewer #2: No

Reviewer #3: Yes

Reviewer #4: Yes

Reviewer #5: Yes

5. Is the manuscript presented in an intelligible fashion and written in standard English?

Reviewer #2: Yes

Reviewer #3: Yes

Reviewer #4: Yes

Reviewer #5: Yes

6. Review Comments to the Author

Reviewer #2: In the revised manusscript, the authors mentioned that three items were found to misfit according to Rasch analysis and mental health was cancelled, the reason should be addressed. Secondly, nine items

Instead of eight ones loaded positively onto the first contrast with a correlation higher than 0.4 in the revised manuscript, why?. Thirdly, they did not test the validity of both visual function and socio-emotional components separately. Finally, some results in revised manuscript is different from the first ones, does it means that the consistency of psychometric properties of the Brazilian Portuguese version of the National Eye Institute Visual Function Questionnaire (NEI VFQ-25) questionnaire in a group of patients with different eye diseases by Rasch analysis is not same?

Reviewer #3: Thank you for putting significant effort into this revision. While the paper still has some weaknesses, you have substantively addressed each of my comments. I don't have futher questions.

Reviewer #4: (No Response)

Reviewer #5: The authors investigate the psychometric properties of the Brazilian Portuguese version of the National Eye Institute Visual Function Questionnaire (NEI VFQ-25) in a group of patients with different eye diseases. This is an important topic, as no study has yet applied Rasch Analysis to the Brazilian Portuguese Version of the NEI VFQ-25. The authors have adressed all comments, so I have no furhter remarks.

7. PLOS authors have the option to publish the peer review history of their article (what does this mean?). If published, this will include your full peer review and any attached files.

Reviewer #2: No

Reviewer #3: No

Reviewer #4: No

Reviewer #5: No

---

## [Author Response · Author response to Decision Letter 1]

16 Nov 2019

We have addressed all comments and suggestions made by reviewers. Details are available in the point-by-point letter response attached.

---

## [Editor Report · Decision Letter 2]

20 Nov 2019

Psychometric Properties of the Portuguese Version of the National Eye Institute Visual Function Questionnaire-25

PONE-D-19-19847R2

Dear Dr. Abe,

We are pleased to inform you that your manuscript has been judged scientifically suitable for publication and will be formally accepted for publication once it complies with all outstanding technical requirements.

With kind regards,

Stefan Hoefer

Academic Editor

PLOS ONE
---

## [Editor Report · Acceptance letter]

2 Dec 2019

PONE-D-19-19847R2 

Psychometric Properties of the Portuguese Version of the National Eye Institute Visual Function Questionnaire-25 

Dear Dr. Abe:

I am pleased to inform you that your manuscript has been deemed suitable for publication in PLOS ONE. Congratulations! Your manuscript is now with our production department. 

With kind regards,

on behalf of

Dr. Stefan Hoefer 

Academic Editor

PLOS ONE